# Placement of Infrastructure for Urban Electromobility: A Sustainable Approach

**Cláudia A. Soares Machado** [1,*], **Harmi Takiya** [2], **Charles Lincoln Kenji Yamamura** [3], **José Alberto Quintanilha** [4] and **Fernando Tobal Berssaneti** [3,*]

1. Department of Transportation Engineering, Polytechnic School, University of São Paulo, São Paulo 05508-070, Brazil
2. Audit Court of the City of São Paulo, São Paulo 04027-000, Brazil; harmi.takiya@tcm.sp.gov.br
3. Department of Production Engineering, Polytechnic School, University of São Paulo, São Paulo 05508-010, Brazil; charles.yamamura@usp.br
4. Institute of Energy and Environment, University of São Paulo, São Paulo 05508-010, Brazil; jaquinta@usp.br
* Correspondence: claudia.machado@usp.br (C.A.S.M.); fernando.berssaneti@usp.br (F.T.B.); Tel.: +55-11-3091-5504 (C.A.S.M.); +55-11-5525-8742 (F.T.B.)

**Abstract:** Over the last few years, electric vehicles (EVs) have turned into viable urban transportation alternatives. Charging infrastructure is an issue, since high investment is needed and there is a lot of demand uncertainty. Seeking to fill gaps in past studies, this investigation proposes a set of procedures to identify the most adequate places for implementing the EV charging infrastructure. In order to identify the most favorable districts for the installation and operation of electric charging infrastructure in São Paulo city, the following public available information was considered: the density of points of interest (POIs), distribution of the average monthly per capita income, and number of daily trips made by transportation mode. The current electric vehicle charging network and most important business corridors were additionally taken into account. The investigation shows that districts with the largest demand for charging stations are located in the central area, where the population also exhibits the highest purchasing power. The charging station location process can be applied to other cities, and it is possible to use additional variables to measure social inequality.

**Keywords:** sustainable transport; charging infrastructure; urban land use; electrical vehicle

---

## 1. Introduction

Over the last two decades, there has been a shift of the global population from rural to urban areas [1]. Such a transition has triggered a significant impact of human trips on global sustainability, making urban areas extremely crowded and polluted and depleting natural resources. However, the movement of people, assets, services, and information is strongly related to economic, social, and environmental development in modern societies, being almost inevitable.

Contemporary society yearns for reductions in the usage of individual motor vehicles, which burn fossil fuels and emit greenhouse gases [2,3]. In the face of such a scenario, electromobility has been deemed a feasible means of sustainable urban mobility [4–10]. An additional benefit of electromobility is low noise pollution [11–13]. The paper [14] ponders that all kinds of urban-oriented vehicles are shifting to electrification: motorized bicycles, two-wheeled vehicles (scooters), automobiles, buses, and delivery vehicles are quickly being developed.

Electric vehicles (EVs) are viable transportation alternatives. In the city of São Paulo, restrictions on the movement of motor vehicles have been adopted, on the grounds of both environmental protection (reduction of greenhouse gas emissions) and mobility improvement (by diminishing the number of

vehicles in the streets). Moreover, government incentives such as tax exemptions and permanent traffic permissions (circulation restrictions are sometimes applied to internal combustion vehicles) encourage the adoption of electric vehicles. With such incentives, electric vehicles have experienced significant adoption and became commonplace in some developed and also emerging countries [15].

In this urban electromobility scenario, [16] claims that a reliable EV-supporting infrastructure is essential, besides a behavioral shift. For [17], although EVs provide significant environmental benefits to society, there are obstacles for their wide adoption, mainly high purchase and maintenance costs and the need for an efficient supporting infrastructure. Charging infrastructure is a major issue. Investment is high, and there is significant uncertainty regarding effective demand. Power grids need to support thousands (eventually millions) of new vehicles extracting tens of kilowatts each. Additionally, batteries need relatively long times for recharging and have limited capacities. Additionally, different customers have a myriad of different needs. The paper [18] points out that, unlike private EVs, electric cabs need relatively short charging times due to continuous operation and distinctive usage patterns. Those factors could inhibit the usage of EVs, as they demand a larger availability of charging stations.

Regarding the methods adopted in this paper, [19] presents a study comparing the impacts of both daytime and nighttime EV charging on electric grids. The evaluation is performed using GIS. The proposed method is highly flexible and easy to apply in other fields of interest. It was clear that night charging was less likely to drive transformer overload than daytime charging. An important inference was the strong influence of pendular travel patterns in the interaction of EV charging with the electric grid. The paper [20] asserts that former charging location models were not adequate to meet EV charging demand and proposes a model based on urban dwellers' trip behavior. It consists of two parts: one for local residents making short trips, using slow-charging installations; another for long distance travelers, who use fast charging. The model aims at both maximum coverage and flow, using mixed integer programming (MIP).

Many researchers are investigating the influence of urban infrastructure in the performance of transport systems [1,21–24]. For [25], the complex and dynamic nature of recharging, the ideal location of stations, and their scheduling led researchers to different optimization algorithms to handle those problems. The paper [25] highlights the importance of EV infrastructure planning. A well-planned and incentivized exploitation of charging stations could lead to a better quality of life, derived from higher EV adoption.

Considering the growing and emerging EV market, significant mobility problems faced by urban agglomeration worldwide, and desire to fill the aforementioned gaps, this study proposes a set of procedures to identify the most adequate places for setting the EV charging infrastructure. The proposed framework was tested and applied in São Paulo city, Brazil. It is versatile and can be easily applied by local and regional authorities, as databases for analysis are publicly available. Besides, the methodology may be used to support the implementation of pilot projects and to guide municipalities, local authorities, and businesses concerned with the advance of electromobility, as well as the implementation of the charging infrastructure needed to assist it.

The remaining sections of this article are organized as follows. Section 2 offers a research and case study overview related to our work. Section 3 presents the study scope and dataset. Section 4 discusses the applied methodology, and Section 5 shows the results and data analysis. Last, the results are discussed and Section 7 concludes.

## 2. Related Work

The purpose of this section is to perform a systematic literature review, providing a replicable, scientific, and transparent process to minimize any potential biases. This procedure identifies key scientific contributions to the field under study [26]. We employed a keyword search approach to identify relevant documents. Several alternative or closely related words were searched, as can be seen in Figure 1. Keywords were looked for in the documents' titles, abstracts, and keywords lists.

Since it is a recent technological innovation, few related studies were found before 2014, but there was a significant increase in publications from 2017, and as in [27], conference papers were included in the analysis to capture the novelty of the subject. As a result, it was possible to identify the most searched keywords. Basic keywords such as "electric vehicles" and "charging infrastructure" appeared before 2017. Keywords related to innovative and disruptive technologies, such as "internet of things" and "vehicle to grid", and environmental issues, such as "sustainability" and "renewable energy resources", started to appear in 2018. In addition, it was possible to detect the countries where these topics are most studied, with the United States playing a prominent role, followed by China, the United Kingdom, Germany, Spain, and Italy.

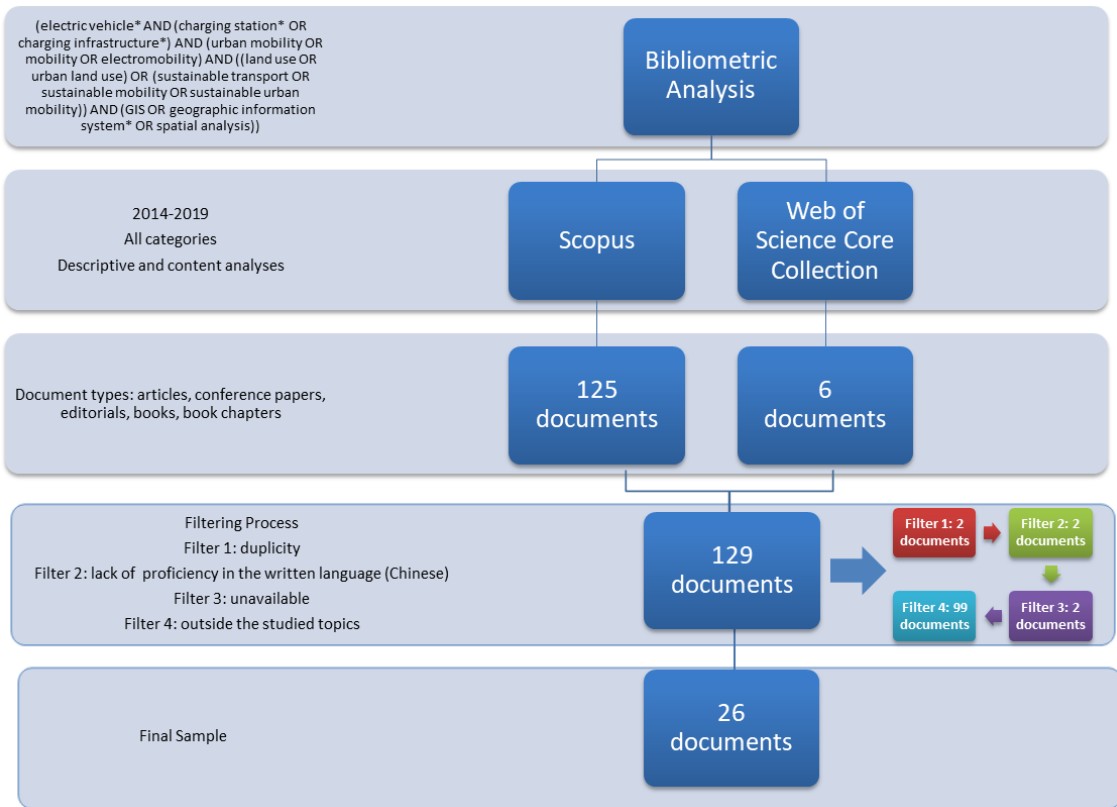

**Figure 1.** Flowchart of the bibliometric study. Source: the authors.

Seven articles were published in 2019; six, in 2018; six, in 2014; five, in 2015; and two, in 2016. Two handbooks and two revision articles were selected as well [14,25,28,29], as can be seen in Figure 1, which synthesizes both the systematic literature review and filtering process. A summary of the contents of the selected articles is as follows:

- Definition of spatial criteria for EV charging station (CS) placement, based on infrastructure, city features, urban planning, and EV user profiles [18,25,30–37].
- Geometric, mathematical, and statistical modeling—such as multicriteria analysis and clustering, geographic information system (GIS), databanks [30,32,35,38–42], and social networks [43].
- Studies on the demand for electric charging, batteries, and the impact of charging stations on electric grids [19,29,32,39,41,42,44–46].
- Several authors make prescriptions for EV charge-point rating, optimizing their allocation. The paper [37], on studying the Northeast of England, used information from the demographic census and socioeconomic data, besides travel patterns, to characterize EV users for new charging installation planning. The paper [30] used a multicriteria method for Point of Interest (POI) allocation, considering the presence of utilities in the Budapest (Hungary) urban area.

- The paper [18] offers a methodology to support decisions on fast-charging station location, targeting cabs—which need short charging times—and taking into account all the stakeholders in the installation process. It yields a spatial database with charging station locations, adjustable by pre-defined criteria. The database may be accessed through GIS by both the municipality and power distribution company.

- The paper [36] claims that the success of plug-in EVs (PEVs) is largely dependent on the accessibility, availability, and quality of the associated infrastructure services (e.g., PEV charging stations in parking lots).

- The paper [29] claims that the charging infrastructure must be set in places with high potential demand, identified by criteria such as the demand density or trip duration. It is noticeable that almost all CS (charging station) location concepts are proposed for urban areas. For [47], infrastructure development is the key factor to increase EV adoption.

## 3. Study Area and Research Data

São Paulo city, the capital of São Paulo state, in the Southeast of Brazil, is the area covered in this study (Figure 2a). It is Brazil's largest city, with a population of 12,176,866 inhabitants (2018) and a demographic density of about eight thousand people per square kilometer. São Paulo city accounts for 11% of the Brazilian gross national product (GNP): USD 194.9 billion in 2015. It has a human development index (HDI) of 0.805 [48]. The municipality is divided into ninety-six districts (Figure 2b), which are the city's smallest administrative spatial units.

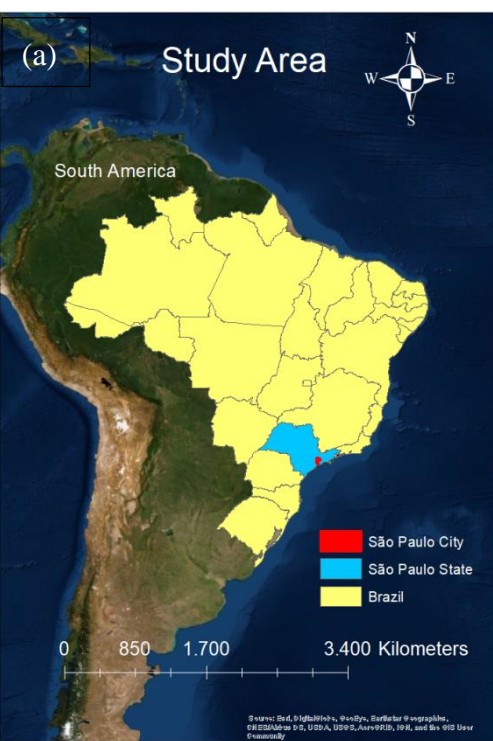
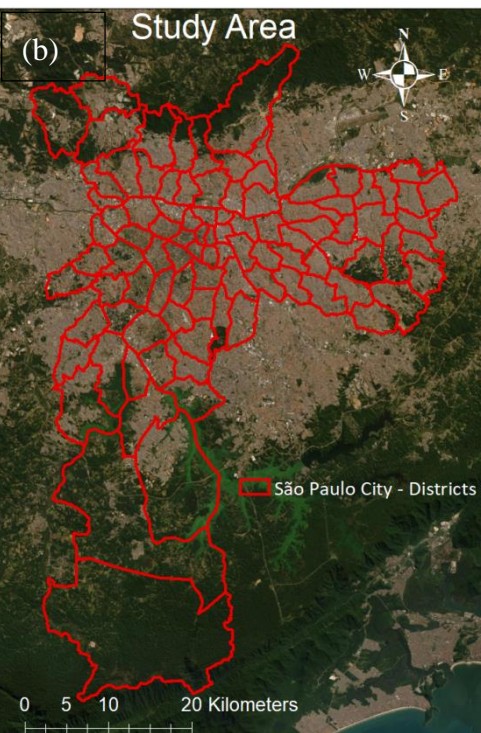

**Figure 2.** Study area: (**2a**) São Paulo City–SP-Brazil; (**2b**) Districts of the City of São Paulo. Source: the authors (software: ArcGIS).

According to the Brazilian National Traffic Department [49], the Brazilian fleet comprised 100,746,553 motor vehicles (automobiles, motorcycles, buses, trucks, etc.) in December 2018. Electric vehicles accounted for only 1% of that fleet (about 10,590 units). São Paulo city had 8,861,208 motor vehicles, and 0.02% of them were electric vehicles (about 1800 units).

The transport systems in the central and western districts provide a high level of accessibility compared to those in other regions of the city, with public services (bus, train, and subway),

bicycle lanes/routes and facilities, a fleet of taxicabs, charter services with buses, microbuses and passenger vans, and some innovative mobility systems (mobility as a service—MaaS) such as Uber and other transportation network companies (TNCs), carsharing, scooter sharing, and bike sharing. Mobility is one of the major problems faced by its residents. Transport is used for commuting to work (18.6 million trips per day) and to school (13.5 million trips a day), according to the 2017 Origin/Destination survey [50]. The rail network is not sufficient to meet existing demand—the subway extends for 89.5 km, and the urban train, 267.5 km. The city bus fleet has fifteen thousand vehicles, which are potential electrification candidates.

The EV market has been growing mainly due to concerns surrounding air pollution. The paper [29] notes a significant 54% increase in worldwide EV sales, compared to those in the years 2016 and 2017. However, [33] deems that those figures are below expectations, due to the lack of adequate charging infrastructure.

## 4. Methods

We adopt a spatial approach, and districts are the research units, as they provide the best publicly available resolution. Georeferenced data were used, which were treated, manipulated, and processed in a GIS (Geographic Information System) environment, for information analysis and integration. Statistical correlation and principal component analysis [51,52] were also used.

In order to identify the most suitable districts for the installation and operation of electric charging infrastructure in São Paulo city, the following public available information was considered for each district: the density of points of interest (POIs, see Table 1), average monthly per capita income, private motor vehicle ownership, and daily trips by transportation mode (data collected by home-based Origin/Destination survey [50]). The current electric vehicle charging network and most important business corridors were also considered. Business corridors refer to the main roads and avenues with large corporate offices and/or intense commerce and service, with heavy vehicle traffic. These routes are located in the most developed and wealthy regions of the city and attract a large volume of daily trips for both work and non-work (e.g., education, health care, shopping, and leisure) purposes.

**Table 1.** Selected points of interest (POIs) and data sources (Source: the authors).

| POI | Source |
|---|---|
| Gas stations | Cadastro Central de Empresas de 2000 (IBGE), elaborated by CEPID-FAPESP/Centro de Estudos da Metrópole (CEM)/Cebrap |
| Private parking lots | |
| Universities and colleges | |
| Supermarkets and grocery shops (with over 100 employees) | |
| Shopping malls | Prefeitura do Município de São Paulo—Dados Abertos (2014) (http://dados.prefeitura.sp.gov.br/dataset/shoppings-centers-no-municipio-de-sao-paulo) |
| Regulated public parking zones ("blue zone") | Prefeitura do Município de São Paulo (2019) (https://www.prefeitura.sp.gov.br/cidade/secretarias/governo/projetos/desestatizacao/estacionamento_rotativo_pago/concessao_estacionamento_rotativo_pago/index.php?p=275579) |

From the analysis of extant information, points of interest (POIs) depict the existence of trip demand, besides the availability of potential charging station setting infrastructure—which could reduce costs in the initial stage. Besides the presence of POIs, social-economic data—monthly per capita income, the number of daily trips, and business corridors—were used, geocodified, and spatialized in the area of study (ninety-six districts), providing an overview of the districts with the highest demand

(business corridors and number of trips) and presence of infrastructure (characterized by POIs) for setting charging stations.

The POIs considered in this paper are gas stations and services, private parking lots, colleges and universities, shopping malls, regulated public parking zones, and supermarkets.

- Gas stations: they are present in significant numbers in all the districts due to the city's large vehicle fleet. Many of them have additional premises for snacks and fast meals and also have enough space for fitting charging stations, as observed by [53].
- Private parking lots: these are also present in all districts, especially in the center area. Some are open for 24 h and permit overnight parking, making it easier to install charging stations. Most of them have enough available space.
- Universities and colleges: many of them are located in the central area. Of course, most attendants are young and sensitive to environmental issues, a target group for electric vehicles, as was also noticed by [37].
- Shopping malls: São Paulo city residents go assiduously to shopping malls (with their shopping spaces and food and leisure facilities) because they provide secure premises with sufficient parking lots, where they can spend from just a few minutes to long hours with their families, especially during the weekends. In the parking spaces, usually, there is enough space for vehicle charging installations.
- Regulated public parking zones: São Paulo city has policies restricting parking on public roads. The so-called "zona azul" (blue zone) comprises parking spots in public roads (streets and parks), allowing a maximum of two-hour parking. The blue zone is a pay-by-use parking scheme, which aims to foster a relay of 41,825 parking lots, to streamline dense road traffic, and to organize urban space, optimizing available parking space. The system provides reserved parking lots for senior citizens, handicapped persons, charter vehicles, and motorcycles. The purpose of restricting and charging for parking on public roads is to discourage the use of private vehicles (especially automobiles), promoting transport modes that are more efficient from social, environmental, and economic perspectives.
- Supermarkets and grocery shops: they are present in all São Paulo districts, and only those with over a hundred employees were considered because those are deemed EV attractors, with a minimum one hour stay in the premises.

The methods do not apply a weighting system, but safeguards were taken. First, we used a correlation matrix to detect the most significant variables. Moreover, a principal component analysis was performed to reduce the dimensionality of the problem, to minimize redundancies, and to eliminate subjectivity in variable selection. From those analyses, we concluded that the most significant variables were (i) the quantity of daily trips according to individual transport mode (automobiles, taxis, motorcycles, and TNCs), (ii) per capita monthly income, (iii) points of interest, and (iv) business corridors. There is not enough information for weighting the relative importance of those variables.

Although the methodology is simple, it is viable, as statistical analysis has backed the selection of variables in the charging station location process. It must also be stressed that the data were standardized to eliminate scaling issues (using values that were comparable among them), to reduce the possibility of bias.

According to [54], the Kernel Density Estimator (KDE) is a popular tool for visualizing data distribution frequently applied in urban studies. The KDE is a non-parametric probability density function estimation method, based on the possibility of analyzing the data without assuming a specific distributional behavior. The papers [55,56], which contributed to the widespread use of this estimator, emphasize the KDE's simplicity and good features and results. The standard KDE density function employs a circular neighborhood as the research area to calculate the density. The KDE characterizes the event location distribution, disregarding its association to values. The result of the KDE application

is a density map, where the value of each image pixel is the relation between event (in this case, POI) concentration per unit area [54].

In the kernel density, the surfaces that surround each event are based on a quadratic (two-dimensional) formula, with the highest values in the center of the continuous surface and gradually decreasing as it moves radially away from the nucleus (radial distance). Kernel (or nucleus) density is calculated using the following mathematical Equation (1) [57]:

$$\hat{\lambda}_\tau(s) = \sum_{i=1}^{n} \frac{1}{\tau^2} I\left(\frac{(s - s_i)}{\tau}\right) \tag{1}$$

where:

$\hat{\lambda}_\tau$ (S) is the intensity of *S*, estimated by kernel density. A quadratic kernel function is assumed, as described in [55] (p. 76, Equation (4.5));

*I* is the probability distribution function and is chosen appropriately to build a continuous surface over the data;

$\tau$ is a smoothing factor known as the "bandwidth" or "radius of influence";

*S* represents any location in the study area;

*Si* is the location of the observed POIs;

*n* represents the number of prediction POIs.

For all the predicted POIs, *I* is approximated by means of kernel density estimation [58,59]. Figure 3 illustrates how the kernel density is calculated.

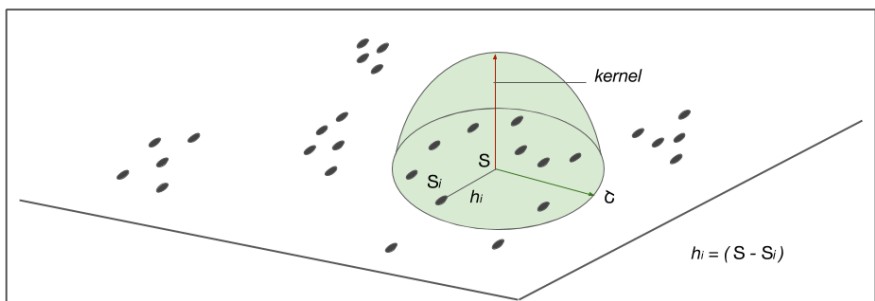

**Figure 3.** Kernel Density Estimator. Source: adapted from [60].

A good review on kernel density estimation for spatial location data can be found in [61]. They investigate the use of kernel density estimation (KDE) methods in detail. Application for the analysis of transportation demand can be found in. [62]. The study [63] uses network kernel density estimation (NetKDE) for detecting traffic accident hot spots. The study [64] uses it to model traffic accidents on residential roads. The paper [65] critically reviews the existing literature on different spatial approaches, including KDE. One of the most cited articles in epidemiology and geography is from [66]. Some classic and basic references including KDE are [57,67–70].

*4.1. Data Description*

A short description of the data sources, geometric aspects, and respective attributes follows.

4.1.1. Punctual Data

POIs are data points scattered over the whole municipality. Each POI was geocodified, and the POI density was calculated using the kernel density estimator—KDE [57]. Table 1 shows the resources used to obtain the POIs (defined in the previous section).

The same POI representation method was used for the existing EV charging infrastructure (source: Associação Brasileira dos Proprietários de Veículos Elétricos Inovadores (ABRAVEI—http://abravei.org/)

and https://www.plugshare.com/, accessed on 3 December 2019. Data for the six aforementioned POIs' defining parameters are publicly and freely available.

### 4.1.2. Linear Data

Business corridors are represented by vectors (lines) and were selected from the information collected in São Paulo city by the research and market intelligence department of Jones Lang LaSalle (JLL) [71]. Business corridors are avenues with high densities of daily trips, especially of private vehicles, for business, work, and school commuting. The analysis considered seven business corridors in São Paulo (Figure 4, Section 5).

### 4.1.3. Polygonal (Area) Data

Ninety-six São Paulo district-delimiting polygons were used, and point, linear, and table data (described next, as attributes linked to district polygons) were associated with them. Information on the daily trips made according transport mode (private automobiles, cabs, motorcycles, and TNCs) and the distribution of the average monthly per capita income of the population [50] (April 2018 average value in U.S. dollars) was laid over the studied area and classified into "very high", "high", "medium", and "low" categories (Figure 5 and Figure 6, Section 5). The Origin/Destination survey from the São Paulo subway company [50] was used as the data source for designing the maps.

## 5. Data Analysis and Results

The density map (for the high, medium, and low categories) was overlaid on the map of the ninety-six São Paulo administrative districts (Figure 4).

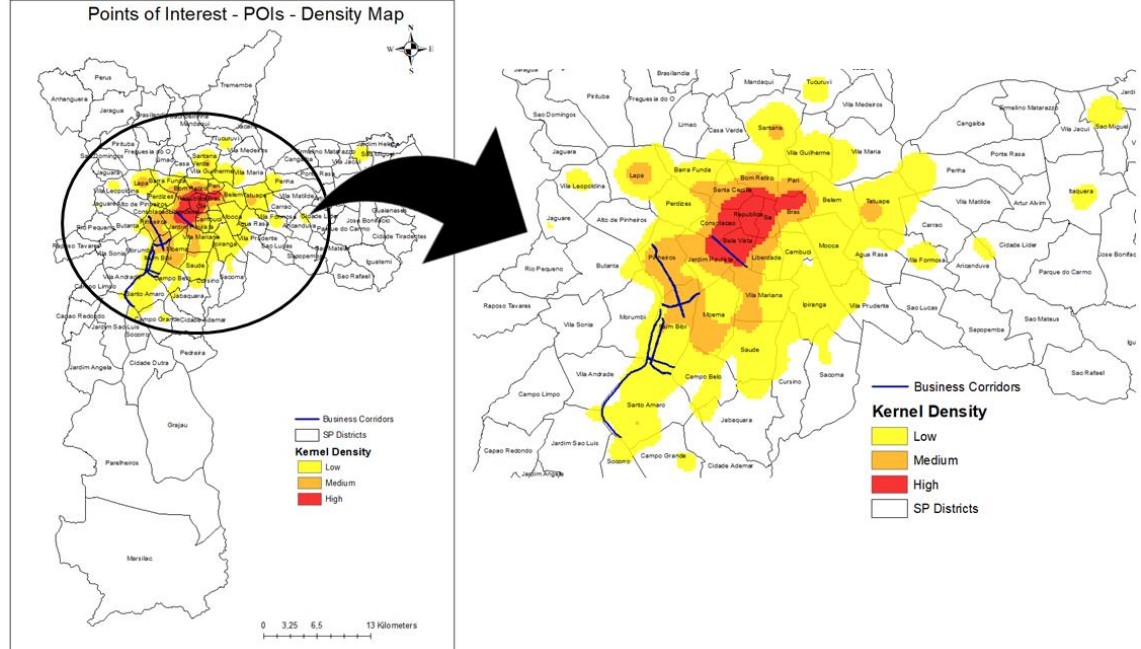

**Figure 4.** POI density hot map, showing spatial POI and business corridor distribution. Source: the authors (software: ArcGIS).

Figures 5 and 6 represent the daily trips (by modality—cars, motorcycles, taxis, and TNCs) and monthly per capita income of the population, respectively. These maps were made with data collected from the origin/destination survey (2017/2018) [50]. The information gathered through home-based interviews (made by household sampling) was extrapolated to the entire study area. Then, the data were spatialized throughout the ninety-six districts of the city of São Paulo. We used the ArcGIS software to draw the maps, using natural groupings inherent in the data (natural breaks) to form the



classes. The separation among the classes was based on breakpoints by picking class breaks or gaps that best grouped similar values and maximized the differences among classes. The features were divided into four classes (in this case labeled "low", "medium", "high", and "very high"), whose boundaries were defined where there were relatively large shifts in data values [72].

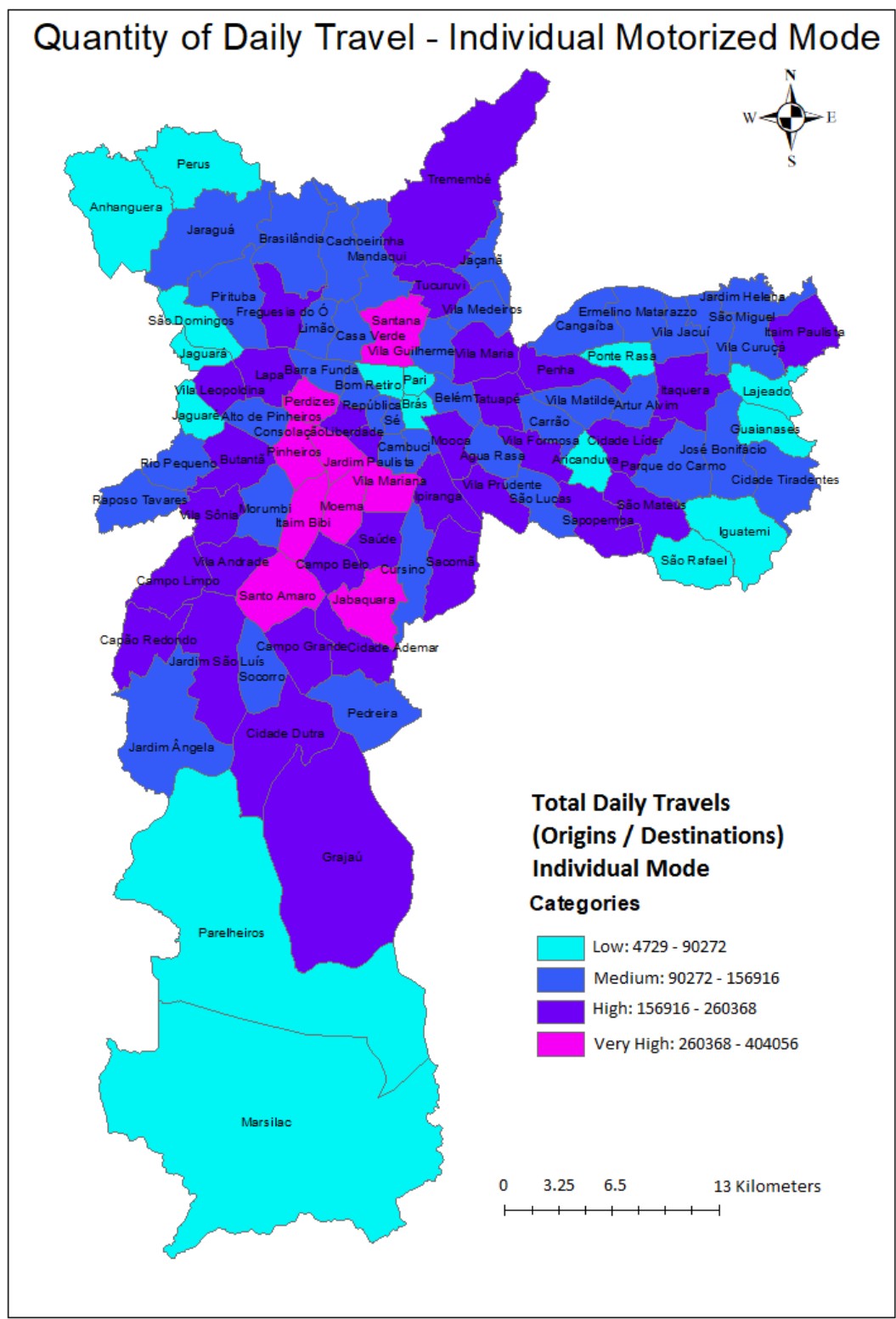

**Figure 5.** Quantity of daily trips by individual transport mode. Source: the authors (software: ArcGIS).

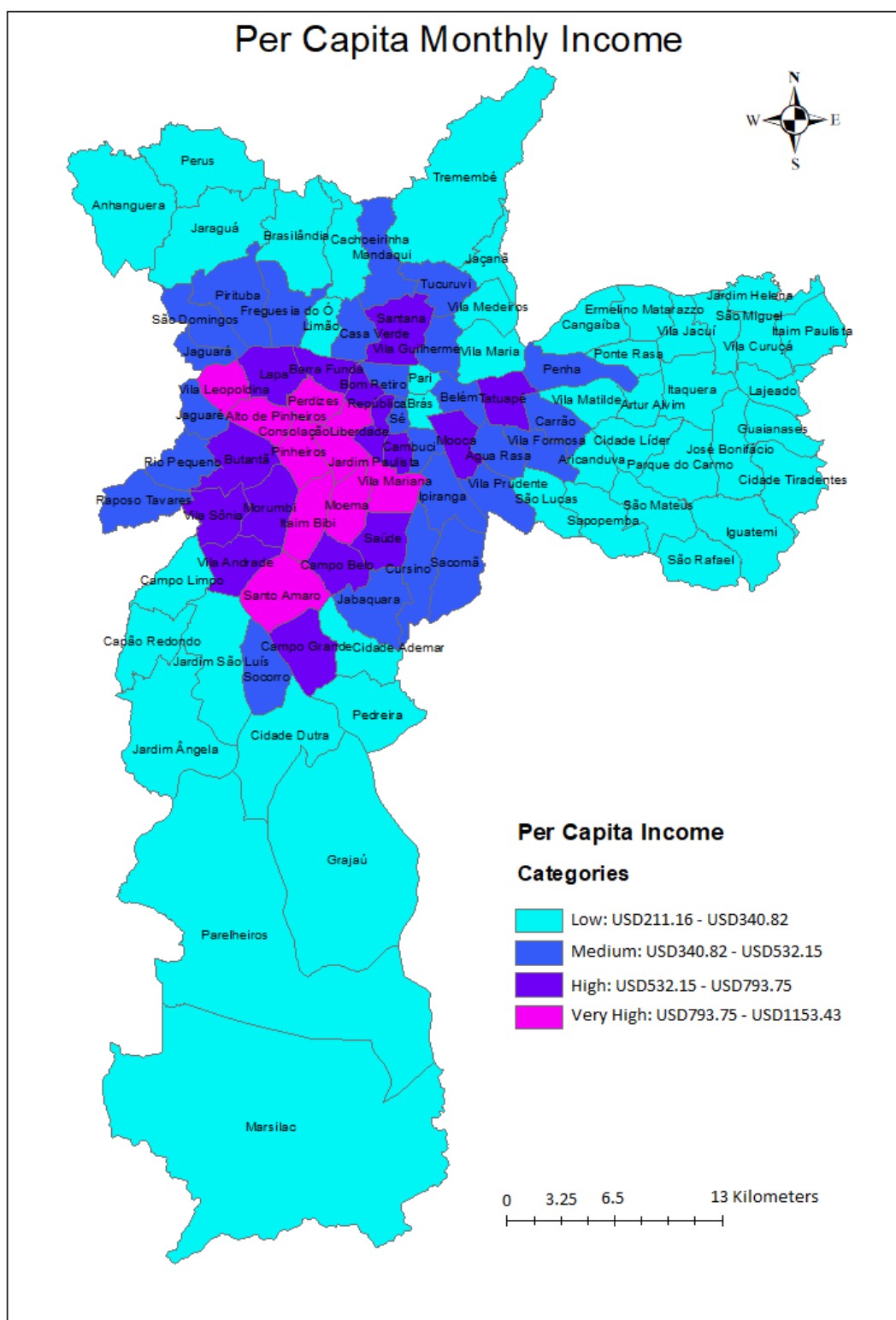

**Figure 6.** Per capita monthly income—April 2018 currency values. Source: the authors (software: ArcGIS).

This type of classification uses the Jenks optimization algorithm to classify attributes. It seeks to minimize the average deviation from the mean for each class while maximizing the deviation from the means of other classes—that is, to reduce the variance within classes and maximize the variance among classes [73,74].

To detect and eliminate possible redundancies, linear correlation and principal component analysis techniques were applied to the average per capita income and private motor vehicle ownership variables. High correlation (0.87) and significant explanation were identified. In line with [34,37], income was chosen for the present work. Daily trips by individual transportation mode (private vehicles, cabs, motorcycles, and TNCs) and average per capita income (April 2018 dollar values) were classified into "very high", "high", "medium", and "low" categories (Figures 5 and 6).

"High" POI density, "very high" daily trip density, and "very high" average income scored "1" each. For the business corridor parameter, a score of 1 was ascribed to districts with one or more corridors. The sums of the scores of the five parameters determined the final district scores (Table 2). Scores of "1" or "0" (presence or absence) were assigned for all the above variables or for business corridor. All parameters received the same weighting, including the six POI categories (gas stations and services, private parking lots, colleges and universities, shopping malls, regulated public parking zones, and supermarkets).

**Table 2.** District final score (source: the authors).

| Ranking | District Name | Total Score |
|---------|---------------|-------------|
| 1° | Jardim Paulista | 4 |
| 2° | Itaim Bibi | 3 |
| 2° | Santo Amaro | 3 |
| 2° | Pinheiros | 3 |
| 2° | Consolação | 3 |
| 2° | Vila Mariana | 3 |
| 7° | Perdizes | 2 |
| 7° | Moema | 2 |
| 7° | Bela Vista | 2 |
| 10° | Alto de Pinheiros | 1 |
| 10° | Vila Leopoldina | 1 |
| 10° | Bom Retiro | 1 |
| 10° | Brás | 1 |
| 10° | Campo Belo | 1 |
| 10° | Jabaquara | 1 |
| 10° | Liberdade | 1 |
| 10° | Pari | 1 |
| 10° | República | 1 |
| 10° | Santa Cecília | 1 |
| 10° | Santana | 1 |
| 10° | Sé | 1 |

The analysis identified twenty-one different sites (administrative districts) in São Paulo as the best candidates for new charging stations. Table 2 shows that the districts with the highest charging station demand are in the central municipality area. They also exhibit the highest purchasing power (scores of 4 and 3) (Table 2).

Currently, São Paulo city has a modest network of electric vehicle charging points, with ninety installed stations, located mainly in strategic locations to attract demand, such as large supermarket chains and shopping malls, as they offer adequate foundations for charging infrastructure. Figure 7 shows the density map of charging stations in São Paulo, which are concentrated in the central and western regions of the city, where workplaces are also concentrated and, for this reason, attract the majority of the daily trips, notably by individual modes of transport. They are also high-income regions. To validate the results, currently existing charging stations are taken as ground truth (Table 3). The five districts with the largest numbers of existing stations validated the nine best-ranking priority districts for new charging stations.

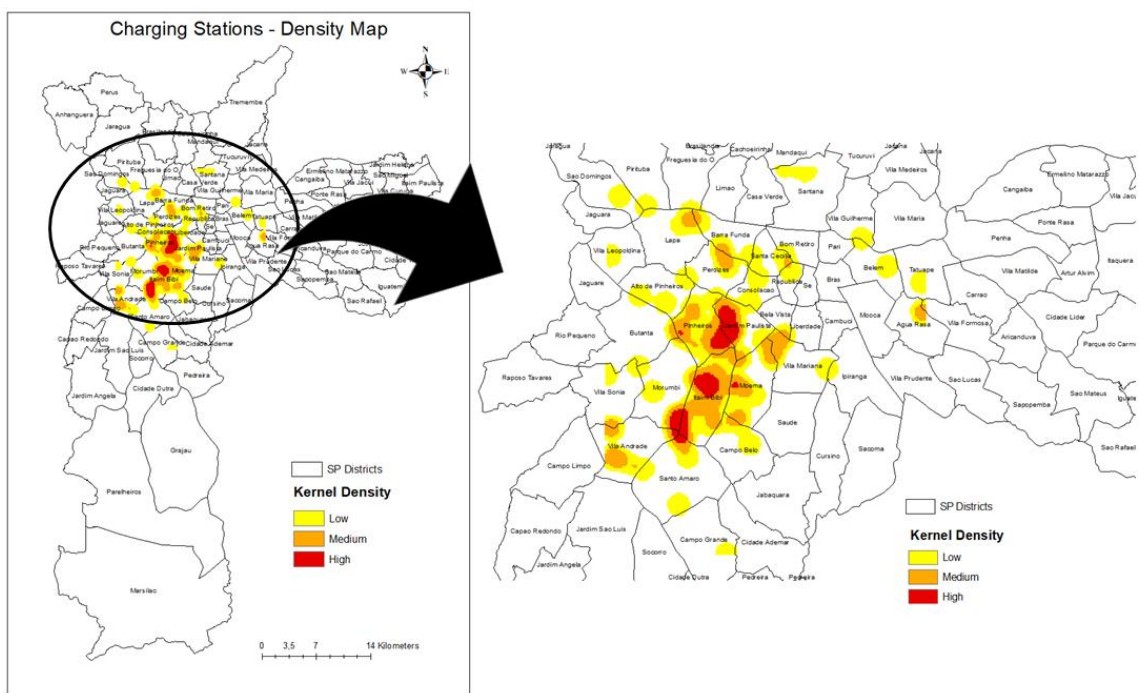

**Figure 7.** Charging infrastructure density hot map, showing spatial distribution of existing charging stations in São Paulo. Source: the authors (software: ArcGIS).

**Table 3.** Ground truth: currently existing charging stations. Source: the authors.

| High Density of Currently Existing Charging Stations |
| :---: |
| Itaim Bibi |
| Jardim Paulista |
| Moema |
| Morumbi |
| Pinheiros |
| Santo Amaro |
| Vila Andrade |

## 6. Discussion

It is important to highlight that the electric vehicle (EV) fleet in Sao Paulo city is still very small (0.02% of the fleet or approximately 1800 vehicles) in comparison to a total fleet of almost nine million vehicles but undoubtedly following the global growth trend. The electric vehicle fleet in Brazil will reach ten million units by 2030. Since São Paulo city accounts for 9% of the national fleet, about 900,000 EVs are expected by 2030. The amount of electricity consumed by transportation (EVs) in Brazil is very small. Specialists believe that the impact of the EV fleet growth should not exceed 1.5% of the country's power load, by 2030 [75].

There are several factors contributing to the EV fleet in the city of São Paulo (also in the whole of Brazil) still being incipient, but the majors ones are the EV acquisition cost—significantly higher than that for combustion engine vehicles—and the unsatisfactory charging network. Therefore, the expansion of the charging infrastructure must be concentrated in places with larger EV circulation (which were detected in this paper) and must be as capillary as possible; that is, they must be homogeneously distributed—in gas stations, supermarkets, shopping malls, colleges, and private and public parking spaces—to encourage the migration to electromobility, presenting it as a reliable proposition by reducing "range anxiety". For that reason, we do not consider one type of POI to be more important than another.

It must be highlighted that changes in both urban and road infrastructure must be made to meet electromobility requirements. For instance, there must be improvements in the mobile internet coverage (4G in São Paulo), the expansion of Wi-Fi networks, the integration of both GPS servers and digital map bases, the development of the Internet and server security protocols, the installation of sensors in traffic lights, bus stops and bicycle/bus/car routes (especially in the bus corridors or BRT—bus rapid transit system), the provision of customer safety, and comfortable street furniture (benches, shelter covers, water fountains, restrooms, information totems, etc.).

The main electromobility users are people keen on technology devices, environmentally aware, and with broad age and background ranges. E-vehicles should provide accessibility to senior citizens, handicapped persons, people with children or infants, tourists who do not speak the native language and/or not familiar with the city, locals seeking to optimize their trips (especially commuting), and so on. Vehicles must be accessible to everyone. For instance, there should be buses with wide doors, low floors, communication systems for announcing stops and other information (in the local language and in English, as a minimum), wheelchair access ramps, wheelchair lodges with stop request commands, reserved seats for handicapped people, front signs with bus line numbers and destinations, etc. Services must provide real-time information on traffic conditions, vehicle location, waiting times, time to destination, routes, coming stops and surroundings (including tourism spots), connections, arrival times at stops, weather, etc. Blind people and wheelchair users must be able to have a free-of-charge companion, ensuring their safety and comfort.

Regarding the methods and obtained results, we understand that the characterization of transport demand is essential for better determining EV charging locations. In this study, daily trips, POIs, business corridors, and average monthly income (which showed a high correlation with motor vehicle ownership) were used. It was noticed that studies with access to a larger amount of georeferenced information—such as details on user profiles, trip needs, and land use, for instance—may yield more precise outputs for charging demand and location. We refer to [30,37], who studied a district of Budapest (Hungary) and the Northeast region of England respectively, with urban features distinctive from those of Sao Paulo and also smaller populations. The study [32] also used the local land use and geomorphological characteristics (such as the proximity to geohazard areas) of Rio de Janeiro city, displaying some variables that express the city's reality. In this monograph, the monthly per capita income variable evidenced the social equality in Sao Paulo, showing significant EV user demand is in high-income population districts.

Therefore, the fitness of data to the reality of the geographic region (town, city, or country) under study is of foremost importance for better defining the EV recharging locations—namely, in reference to the EV customer profile, e.g., social economic data, demand for trips (work, school, and leisure), and traveling distances (short and long). The quantity of data per se is not of paramount importance, provided they are representative of the sampled geographical region. However, it is very important that data are publicly available, georeferenced, and reliable.

In this paper, the polygon that represents the smallest municipality administrative unity—the district—was used. Ninety-six districts, with diverse areas, are distributed over 1509 km$^2$. Although the studied areas are larger than those in the studies [30,37], each São Paulo district has a larger impact, as the city administration refers to districts.

Five of final districts pointed out by the research, with large scores, are in the current stations list. Station location was defined by commercial interest: available space and local access by medium- and high-income people. All seven charging stations are supported by automobile manufacturing firms. Two spots (Morumbi and Vila Andrade) are located around large upscale shopping malls but in regions with enormous social contrast: many low-income people living mainly in slums (such as the Paraisopolis community, with more than 100,000 inhabitants—the second-largest slum in the city) and some very high-income condominiums. For this reason, these two districts did not exhibit high scores. This large local social inequality was not detected by our methodology.

## 7. Conclusions

EVs are gradually becoming a viable and widespread urban transportation alternative. However, charging infrastructure is an issue, as significant investments are needed and there is a lot of uncertainty about demand, especially in developing countries. Moreover, there is a lack of reliable, good-quality information.

This article addresses a gap in the literature on the location of EV charging stations. It started with a systematic literature review, relating EVs, charging stations, and spatial distribution. The number of publications in the field has increased substantially from 2018 to the present. The United States, China, the United Kingdom, Germany, Spain, and Italy are the leaders in the production of scientific articles related to that topic. The considered bibliographic portfolio shows the most relevant journals and publications in the field related to this paper. The search comprised conference papers, articles, editorials, books, and book chapters and identified the most searched keywords. A shift in the main set of keywords was detected in 2018.

The present study has achieved its proposed objective—to identify the best potential EV charging point locations in São Paulo city, based on social-economic and transport demand data.

The paper presents a set of procedures that was developed to identify the most favorable districts for the installation and operation of electric charging infrastructure in the city of Sao Paulo, using public available information and spatial analysis. The spatial unit adopted, the city administrative district, was chosen since it was the scale of public information available, in contrast with the disposal information from cities in the first world, where they could adopt blocks or lanes as spatial units. Besides these limitations, the main objective—identifying the best potential EV charging point locations in São Paulo city—was achieved, according to socio-economic and transport demand data, as attributes of city districts. In this sense, socio-economic data lays out the enormous social inequality, which imposes geographic and financial boundaries across the city.

Aiming to detect and eliminate possible redundancies among the initial chosen variables, linear correlation and principal component analysis techniques were applied to the average monthly per capita income and rate of ownership of private motor vehicles variables, showing that only one of these variables was enough to explain the socio-economic disparity.

These characteristics, peculiar to developing countries, enable the method to be applied to other cities where similar public georeferenced data are available, since the proposed framework is versatile and easy to be applied by local and regional authorities. Besides, the methodology may be used to support the implementation of pilot projects and to guide municipalities, local authorities, and businesses concerned with the advance of electromobility, as well as the implementation of the charging infrastructure needed to assist it.

Besides the locations of the currently existing charging stations being determined according to the EV manufacturers' criteria, without the consideration of any kind of geographic and social characteristics of the city, those locations were compared with the locations determined by the procedures and were used to validate the results. The currently existing charging stations were taken as ground truth and compared with the used-procedure results. The two different sites detected were in regions where the socio-economic contrast is enormous (shopping malls located in the neighborhood of big slums).

Future work could consider surveys (such as origin/destination) of the used POIs for determining more accurate estimates of the numbers of trips and other attributes (preferential trip periods and user profiles, for example) and recent digital images from satellites or unmanned aerial vehicles (drones) to design spatial units better than city administrative districts, improving the results scale.

**Author Contributions:** Data curation, H.T.; formal analysis, C.A.S.M. and J.A.Q.; investigation, C.A.S.M. and H.T.; methodology, J.A.Q. and F.T.B.; project administration, F.T.B.; resources, F.T.B.; writing—original draft, C.A.S.M.; writing—review and editing, C.L.K.Y., J.A.Q., and F.T.B. All authors have read and agreed to the published version of the manuscript.

**Funding:** This study was financed in part by the Coordenação de Aperfeiçoamento de Pessoal de Nível Superior-Brasil (CAPES)—Finance Code 001—and National Council for Scientific and Technological Development (CNPq), Brasilia, Brazil—Finance Code 305987/2018-6.

**Acknowledgments:** The authors thank the Polytechnic School and Institute of Energy and Environment of the University of São Paulo, the CAPES (Higher Education Personnel Improvement Coordination) grant number 16.1.2226.3.8, and Fundação Carlos Alberto Vanzolini (FCAV) for financial support. We also thank Mauricio Faria and Alexandre Cordeiro, of the Sao Paulo City Audit Office, for the support for the development of this research, and Mariana de Oliveira Lage for processing the information at the initial stages of this investigation. Special thanks go to the reviewers of the Sustainability periodical for their valuable contribution to the publication of this study.

**Conflicts of Interest:** The authors declare no conflict of interest.

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
