# Peer review of "Placement of Infrastructure for Urban Electromobility: A Sustainable Approach"

_sustainability, doi:10.3390/su12166324_

Round 1
Reviewer 1 Report
The authors have made a great effort in designing an analysis methodology to identify the optimal locations for the placement of infrastructure for electromobility. However, it is necessary to correctly present the state of the art on the subject (introduction) as well as the methodology used and especially the discussion of the data and conclusions. Right now, the discussion and conclusions are together and I think they should be separated. I think it is important to compare the data obtained with those of other cities in other contexts in order to determine the common points or disparities. In the same way, you define a methodology that you say can be replicated in other cities but in the discussion and conclusions you do not clearly indicate where your methodology is better than the previously published ones (what are the positive or negative factors that make yours better than previous ones?)
In general, you should adapt the document to the publication standards of the journal. The format for citations and references is not correct. The format of Figures and Tables is not correct and several sections that are mandatory in the journal are missing.

Author Response
Dear Reviewer #1
Thank you very much for your time and valuable comments and suggestions provided. In this revised version, corrections and explanation have been made to clarify the issues pointed in this paper. The revisions were highlighted using the “track changes” mode in Microsoft Word, so the changes are easily visible. In addition, during the review process, the authors provided a grammatical, syntactic, and orthographic revision of English language.
Comments:
"The authors have made a great effort in designing an analysis methodology to identify the optimal locations for the placement of infrastructure on electromobility. However, it is necessary to correctly present the state-of-the-art on the subject (introduction) as well as the methodology used and especially the discussion of the data and conclusions”.
Response:
The authors agree with the reviewer. Section 1 (Introduction) was expanded to better present the state-of-the art of the topics addressed in this paper – page 2, lines 57-67.
The section 4 (methods) was restructured. A justification about the choice of POIs was incorporated in the manuscript – page 8, lines 235-241. Also, a new description of the elements of equation (1) was included in the text – pages 9 and 10 (lines 315-325). In addition, a figure was introduced (now figure 3 – page 10) to exemplify the calculation of KDE by equation (1). Finally, a new paragraph was added to the text in order to address other studies applying KDE to spatial location data (as described in equation 1). This excerpt can be seen on page 10, lines 330-338.
Similarly, section 5 (data analysis and results) was improved. A description of how the former figures 6 and 7 (now figures 4 and 5) were obtained (page 11, lines 379-395), and an explanation on how the scores were assigned (page 15, lines 419-425) were provided.
The section 6 (discussion) was expanded to meet the reviewer's request – page 17, lines 479-498. The same was provided in the new section 7 (conclusion) – pages 17 and 18.
“Right now, the discussion and conclusions are together and I think they should be separated”.
Response:
The authors agree with the reviewer. Former section 6 (Discussion and Conclusion) was split into tw sections: new section 6 (discussion – pages 16 and 17) e section 7 (conclusion – pages 17 and 18).
“I think it is important to compare the data obtained with those of other cities in other contexts in order to determine the common points or disparities. ln the same way, you define a methodology that you say can be replicated in other cities but in the discussion and conclusions you do not clearly indicate where your methodology is better than the previously published ones (what are the positive or negative factors that make yours better than previous ones?”
Response:
The authors agree with the reviewer. In section 6 (discussion), the authors added two paragraphs (page 17, lines 479-498), which provide the discussions suggested, comparing proposed methods with those in some published documents. It is worth mentioning we do not intend the proposed methodology to be “superior” to previously published ones - applied to different geographical contexts. We deem both the used method and chosen data are the most adequate to the city of Sao Paulo, as they portray its reality (including social inequality). On the other hand, replicability refers to using the technique with similar (especially publicly available) data. With due urban and social context analysis, the method is replicable and can yield the most adequate spots for placing EV charging stations.
“ln general, you should adapt the document to the publication standards of the journal. The format for citations and references is not correct. The format of Figures and Tables is not correct and several sections that are mandatory in the journal are missing”.
Response:
The authors agree with the reviewer, on bibliographical citations. However, we could not identify the referred detailed standards on figures, tables, and section structures. On paper approval, we will advance the format changes requested by the journal’s editorial staff.
Remarks:
- The other changes made to the text meet the requests of the reviewer #2, and the English editing.
- On manuscript revised version the pages and lines indicated correspond to the text when “Track Changes” is turned on; option “All markup”.
Reviewer 2 Report
Dear authors:
Section 2 I consider should be decreased. Figure 2 seems to me to summarize the procedure and for my way of seeing it is sufficient. The figure, 1,3,4, I consider should be eliminated.
In the Methods section there is a literature review that could be integrated into section 2.
The elements of equation 1 are not fully described. It is recommended to indicate more studies in which Equation 1 was used. It is recommended that it be indicated in more detail how figures 6 and 7 were obtained (remember that all research must be replicable) . It is required to justify the choice of POIs.
It is recommended to include in more detail how the scores are assigned. I understand there is only a 0 or 1 rating. This could be a problem since all POIs are assumed to have equal weight. For example, there may be a district that has a very high POI density, in some attribute more important than another. I consider that in this sense there is no solid methodology, so it would be appropriate for me to look for another, deeper way to qualify POIs.
A comprehensive review of the state of the art has been presented, but no discussion. It is understood as a discussion that the results obtained are contrasted with similar research. In this sense, the results are not validated.
With the above, although what the manuscript tries to demonstrate seems appropriate to me, I consider that the methodology is too elementary, therefore the manuscript such as this is not suitable for publication in a journal.
Author Response
Dear Reviewer #2
Thank you very much for your time and valuable comments and suggestions provided. In this revised version, corrections and explanations have been made to address the issues in this paper. The revisions were highlighted using the “Track Changes” function in Microsoft Word, so the changes are easily visible. In addition, during the review process, the authors provided a grammatical, syntactic and orthographic revision of English language.
Comments:
“Section 2 I consider should be decreased. Figure 2 seems to me to summarize the procedure and for my way of seeing it is sufficient. The figure, 1,3,4, I consider should be eliminated”.
Response:
The authors agree with the reviewer. Section 2 was decreased and restructured. Figures 1, 3, and 4 were deleted from the text. On pages 2 and 3, the excerpt between lines 95-98 was eliminated, because it is explained by figure 2 (now, figure 1). For the same reason, part of the line 100 (page 3), and lines 144-145 (page 4); and lines 135-136 (page 4) were eliminated as well. The paragraphs between lines 119-134 (Page 3) were transferred to the end of the section, in order to improve reading fluidity (now lines 152-176 – pages 5 and 6).
“ln the Methods section there is a literature review that could be integrated into section 2”
Response:
The authors agree with the reviewer. Among the five referred bibliographical citations in section 4 (Methods), three were incorporated to section 2 - (Morro-Mello et al., 2019; Su & Hu, 2015; Pagany, Ramirez Camargo & Dorner, 2019) – page 6, lines 177-189). Two other articles (Heymann et al., 2017; Sun et al., 2018) were allocated to section 1 (introduction), giving better consistency to specific references in paper (page 2, lines 57-67). Therefore, on page 8, lines 242-269 were eliminated from section 4 (methods) and reallocated as described below.
“The elements of equation 1 are not fully described”.
Response:
The authors agree with the referee and new descriptions of elements of equation (1) were included in the text – pages 9 and 10 (lines 315-325). In addition, a new figure was introduced (figure 3 – page 10) to exemplify the calculation of KDE by equation (1).
“It is recommended to indicate more studies in which Equation 1 was used”.
Response:
The authors agree with the referee and a new paragraph was added to the text in order to address studies that apply KDE for spatial location data (as described in equation 1). This excerpt can be seen on page 10, lines 330-338.
“It is recommended that it be indicated in more detail how figures 6 and 7 were obtained (remember that all research must be replicable)”.
Response:
The authors agree with the reviewer. An explanation about these figures (now figures 4 and 5 – pages 12 and 13) was introduced in the text; on pages 11 and 12 (lines 379-395).
“lt is required to justify the choice of POls”
Response:
The authors agree with the reviewer. On section 4 (Methods), the authors added a new paragraph (page 8, lines 235-241), justifying POI selection.
“lt is recommended to include in more detail how the scores are assigned. I understand there is only a O or 1 rating. This could be a problem since all POls are assumed to have equal weight. For example, there may be a district that has a very high POI density, in some attribute more important than another. I consider that in this sense there is no solid methodology, so it would be appropriate for me to look for another, deeper way to qualify POls”.
Response
The authors partially agree with the reviewer. The authors have added a paragraph on item 5, Data Analysis and Results (page 15, lines 419-425), detailing the scoring process.
About weighting, the authors have analyzed data for each district, and adopted a binary criterion (presence = 1; absence = 0). For example, districts with business corridors (large avenues) were scored “1”; the same for “very high” monthly per capita income and total amount of daily trips. About POIs, the six parameter categories (gas stations and services, private parking lots, colleges and universities, shopping malls, regulated public parking zones, and supermarkets) were considered as having the same importance (weight), to avoid error from lack of more detailed information, like daily flow of vehicles in parking lots, shopping malls, and supermarkets, and trip distances to those services. They were chosen as points-of-interest because they can potentially provide adequate infrastructure for placing EV recharging points, besides being strong demand attractors.
It is worth mentioning several authors, e.g. Csiszar et al. (2019), Pagany, Ramirez Camargo & Dorner (2019), Namdeo et al. (2014), point demand, service attraction and social-economic data - mentioned in this work - as parameters for defining the best location of EV charging points.
“A comprehensive review of the state of the art has been presented, but no discussion. It is understood as a discussion that the results obtained are contrasted with similar research. In this sense, the results are not validated”.
Response:
The authors partially agree with the reviewer. In section 6 (discussion), the authors added two new paragraphs (page 17, lines 479-498), which provide the suggested discussions, comparing proposed methods with those in some published documents. It is worth mentioning we do not intend the proposed methodology to be “superior” to previously published ones, which are applied to different geographical contexts. We deem both the used method and chosen data are the most adequate for the city of Sao Paulo, as they portray its reality, including social inequality. On the other hand, replicability refers to using the technique with similar (especially publicly available) data. With due urban and social context analysis, the method is replicable and can yield the most adequate spots for placing EV charging stations.
We disagree on the lack of validation. The methodology was verified by extant charging station density maps. Station location was defined by commercial interest: available space and access to medium and high incomes. Obtained results concern the municipality of São Paulo. All seven charging stations are supported by automobile makers (that information was not previously mentioned). Five of the districts pointed out in the research, with large scores, are present in the current stations list. Among the seven districts previously selected, two (“Morumbi” and “Vila Andrade”) are placed around large upscale shopping malls but are located in regions with enormous social contrast: many people with very low income, living mainly in slums (as the Paraisópolis community, with more than 100,000 inhabitants - the 2nd largest slum in the city), and some very high income condominiums. For this reason, these two districts do not exhibit high scores. This very large local social inequality was not detected in our methodology.
Remarks:
- The other changes made to the text meet the requests of the reviewer #1, and the English editing.
- On manuscript revised version the pages and lines indicated correspond to the text when “Track Changes” is turned on; option “All markup”.
Round 2
Reviewer 1 Report
The authors have made a great effort to address all the comments defined in the review process. However, in my opinion, the conclusions of the document are not correctly drafted. Again, It is a summary but It does not determine what the main findings of the research are.
In addition, authors should modify the format of the document to conform to that of the journal. By not doing so, the authors demonstrate a lack of interest in publishing the document in this particular journal. They should be meticulous about such details. For example, the Contributions section has yet to be incorporated into the document.
On the information provided in the discussion (L 377-388), it is not enough to include a paragraph with information. Authors should DISCUSS THE DATA. It should not be the reader who draws conclusions, but the authors who define them.For example, in other published studies, the number of georeferenced data was higher. Is this a positive or negative thing? What are the advantages and disadvantages? How much more data (in the order of 3/5/6 times more)? All these details help to understand the discussion of the results.
Author Response
Dear Reviewer #1
Once again, the authors would like to thank you for your time and valuable comments and suggestions provided. In this revised version, corrections and explanations have been made to address the issues in this paper. In addition, during the review process, the authors provided a grammatical, syntactic and orthographic revision of English language.
Comments:
“The authors have made a great effort to address all the comments defined in the review process. However, in my opinion, the conclusions of the document are not correctly drafted. Again, it is a summary but it does not determine what the main findings of the research are”.
Response
Besides the authors had prepared a new version of the document attending to the other Referee, where the item Discussion and Results was divided in two new individualized items and including some others aspects, we agree with the Referee and re-write the Conclusions (pages 14 and 15).
“In addition, authors should modify the format of the document to conform to that of the journal. By not doing so, the authors demonstrate a lack of interest in publishing the document in this particular journal. They should be meticulous about such details. For example, the Contributions section has yet to be incorporated into the document”.
Response
The authors agree with the reviewer. The paper format was modified according to journal standards. The missing mandatory sections were included (pages 15 and 16 – lines 486-505); as well the references and citations format (pages 16-19). Throughout the text the new format of citations was highlighted with the color blue.
“On the information provided in the discussion (L 377-388), it is not enough to include a paragraph with information. Authors should DISCUSS THE DATA. It should not be the reader who draws conclusions, but the authors who define them. For example, in other published studies, the number of georeferenced data was higher. Is this a positive or negative thing? What are the advantages and disadvantages? How much more data (in the order of 3/5/6 times more)? All these details help to understand the discussion of the results.”
Response
The authors agree with the reviewer. A more detailed explanation is provided in section 4 (methods) in page 6 - lines 208 to 218, and the discussion topic (section 6), page 13 - lines 350-352 and lines 357 to 365; and page 14 – lines 397-406.
Remark:
- On manuscript revised version the pages and lines indicated correspond to the text when “Track Changes” is turned on; option “All markup”.
Reviewer 2 Report
Dear Authors.
Thank you for receiving my comments and using them to improve your manuscript. Despite the fact that the requested correction has been made, in my opinion it is very elementary to assume that all the parameters have the same weight, it was possible to carry out a weight assessment, which would allow to eliminate subjectivity. There are several methodologies that would allow evaluating these weights, which include, for example, surveys. In a city like São Paulo, it would be appropriate to use these assessments.
That I consider to be the drawback of its methodology. Likewise, the format requirements of the publication are not being followed, so I suggest that the manuscript be improved before publication.
Author Response
Dear Reviewer #2
Once again, the authors would like to thank you for your time and valuable comments and suggestions provided. In this revised version, corrections and explanations have been made to address the issues in this paper. In addition, during the review process, the authors provided a grammatical, syntactic and orthographic revision of English language.
Comments:
“Thank you for receiving my comments and using them to improve your manuscript. Despite the fact that the requested correction has been made, in my opinion it is very elementary to assume that all the parameters have the same weight, it was possible to carry out a weight assessment, which would allow to eliminate subjectivity. There are several methodologies that would allow evaluating these weights, which include, for example, surveys. In a city like São Paulo, it would be appropriate to use these assessments. That I consider to be the drawback of its methodology”.
Response
The authors agree with the reviewer that a weight assessment would consolidate the proposed methodology. However, there is no data currently available in São Paulo to conduct weight assessment.
From the beginning, the authors have assessed the appropriateness of a weighting system. We have searched several potential sources (public and free datasets), but to no avail. The authors also considered generating this information through field research. However, the Covid-19 pandemic frustrated such expectations. For instance, the official bodies of the São Paulo city hall have been virtually inaccessible since March 2020, assistance being restricted to emergency situations. Besides, funding, time and a considerable staff would be necessary to conduct such surveys, and we do not possess such resources. Academic literature surveys and methods applying weight assessment, - with variables and an urban design similar to ours - were also searched, but we were unsuccessful.
Therefore, the authors decided to do not change the weighting system, but safeguards were taken. First, we used a correlation matrix to detect the most significant variables. Moreover, a principal component analysis was performed to reduce the dimensionality of the problem, to minimize redundancies, and to eliminate subjectivity in variable selection. From those analyses, we concluded the most significant variables were (i) quantity of daily trips by individual transport mode (automobiles, taxis, motorcycles, and TNCs); (ii) per capita monthly income; (iii) points of interest; and (iv) business corridors. We do not have enough information for weighting the relative importance of those variables.
Although the methodology is simple, it is viable, as statistical analysis has backed the selection of variables in the charging station location process. It must also be stressed data was standardized to eliminate scaling issues (using values that are comparable among them), to reduce the possibility of bias. This explanation was included in the manuscript (page 6, lines 219-229).
It is important to highlight the electric vehicle (EV) fleet in Sao Paulo is still very small (0.02% of the fleet, or approximately 1,800 vehicles) in comparison to a total fleet of almost nine million vehicles, but undoubtedly following the global growth trend. That is due to several factors, but mainly EV acquisition cost, significantly higher than for combustion engine vehicles, and unsatisfactory charging network. Therefore, the expansion of the charging infrastructure must concentrate in places with larger EV circulation (which were detected in our methodology), and must be as capillary as possible; that is, they must be homogeneously distributed, in fuel stations, supermarkets, shopping malls, colleges, private and public parking spaces, to encourage the migration to electromobility, presenting it as a reliable proposition by reducing “range anxiety”. For that reason, we do not consider one type of POI more important than another. This explanation was included in the manuscript (page 13, lines 365-367, and lines 372-380).
“Likewise, the format requirements of the publication are not being followed, so I suggest that the manuscript be improved before publication”.
Response
The authors agree with the reviewer. The bibliographical citations do not follow the journal standard. This and other issues regarding format will be solved during the editorial process. On paper approval, we will advance the format changes requested by the journal’s editorial staff.
Remark:
- On manuscript revised version the pages and lines indicated correspond to the text when “Track Changes” is turned on; option “All markup”.
Best Regards